# Efficient Nanocrystal Photovoltaics with PTAA as Hole Transport Layer

**DOI:** 10.3390/nano12173067

**Published:** 2022-09-03

**Authors:** Ao Xu, Qichuan Huang, Kaiying Luo, Donghuan Qin, Wei Xu, Dan Wang, Lintao Hou

**Affiliations:** 1State Key Laboratory of Luminescent Materials & Devices, Institute of Polymer Optoelectronic Materials & Devices, South China University of Technology, Guangzhou 510640, China; 2Guangdong Provincial Key Laboratory of Optical Fiber Sensing and Communications, Guangzhou Key Laboratory of Vacuum Coating Technologies and New Energy Materials, Siyuan Laboratory, Department of Physics, Jinan University, Guangzhou 510632, China

**Keywords:** CdTe, nanocrystal, solar cells, hole transfer materials, PTAA

## Abstract

The power conversion efficiency (PCE) of solution-processed CdTe nanocrystals (NCs) solar cells has been significantly promoted in recent years due to the optimization of device design by advanced interface engineering techniques. However, further development of CdTe NC solar cells is still limited by the low open-circuit voltage (*V*_oc_) (mostly in range of 0.5–0.7 V), which is mainly attributed to the charge recombination at the CdTe/electrode interface. Herein, we demonstrate a high-efficiency CdTe NCs solar cell by using organic polymer poly[bis(4–phenyl)(2,4,6–trimethylphenyl)amine] (PTAA) as the hole transport layer (HTL) to decrease the interface recombination and enhance the *V*_oc_. The solar cell with the architecture of ITO/ZnO/CdS/CdSe/CdTe/PTAA/Au was fabricated via a layer-by-layer solution process. Experimental results show that PTAA offers better back contact for reducing interface resistance than the device without HTL. It is found that a dipole layer is produced between the CdTe NC thin film and the back contact electrode; thus the built–in electric field (*V*_bi_) is reinforced, allowing more efficient carrier separation. By introducing the PTAA HTL in the device, the open–circuit voltage, short-circuit current density and the fill factor are simultaneously improved, leading to a high PCE of 6.95%, which is increased by 30% compared to that of the control device without HTL (5.3%). This work suggests that the widely used PTAA is preferred as the excellent HTL for achieving highly efficient CdTe NC solar cells.

## 1. Introduction

Recently, solution-processed inorganic thin film solar cells with semiconductor nanocrystals (NCs) as the active layer have attracted much attention for their low-cost photovoltaic applications [1,2,3]. Compared to the traditional inorganic thin film prepared by thermal vacuum evaporation, the low ingredient and power consumption along with excellent compatibility with an industrial roll-to-roll fabrication process makes solution-processed NC solar cells more promising for large scale manufacturing [4,5,6]. Among all kinds of NC solar cells, CdTe NC solar cells have received intensive research interest because of their ideal bandgap (~1.5 eV), endurance to ambient conditions and mature synthesis process for high quality NC production [7,8,9,10]. A power conversion efficiency (PCE) of ~12% [11] has been obtained with a conventional device structure and 9.2% with an inverted device architecture of ITO/ZnO/CdS/CdSe/CdTe/P-TPA/Au using solution-processed CdTe NC thin film as the active layer [12]. To further improve the efficiency of CdTe NC solar cells, the device structure design and interface optimization are still challenging because of the low doped concentration and high valence band of CdTe NC film [13]. Traditionally, depositing a thin layer of Cu or Cu salt on the CdTe NC thin film can increase the hole density and reduce contact resistance, which have been applied successfully to CdTe thin film solar cells prepared by vacuum technique [14,15]. However, it is difficult to steadily obtain Cu doped film for the crystal boundaries in CdTe NC thin film. A large amount of Cu can diffuse easily along the CdTe NC grain boundary into the cell junction and form recombination centers and shunt paths, limiting the cell performance [15]. In the case of solution processed inorganic thin film solar cells, inserting a hole transport/interfacial layer between the active layer and the contact electrode has been widely adopted [16]. Thanks to the good conductivities, suitable band alignment and low reactivity, the small molecule 2, 2′,7,7′–tetrakis (N,N–di–p–methoxy–phenyl-amine)–9,9′–spirobifluorene (Spiro–OMeTAD) or the conjugated polymer poly[bis(4–phenyl)(2,4,6–trimethylphenyl)amine] (PTAA), have been widely used as the hole transport layer (HTL) to build high-efficiency perovskite solar cells (PSC) [17,18,19,20,21,22]. By doping bis (trifluoromethane) sulfonamide lithium salt (Li-TFSI) and 4-tert-butylpyridine (t-BP) into HTL, the hole mobility, conductivity and carriers extraction efficiency of PSC are further improved [23,24,25]. As the CdTe NC has similar electronic/chemical properties as the perovskite, the performance of CdTe NC solar cells may be improved by inserting an organic HTL between the CdTe NC and the contact electrode. Although PTAA has been used successfully in PSCs as HTL, it has not been explored for CdTe NC solar cells. In previous reports, the effects of the interactions between HTL and CdTe NC films have been studied. The type of molecule, precursor concentration and post-treatment of HTL are crucial to device performance. It has been demonstrated that only suitable HTLs can reduce carrier recombination and increase device performance [26]. Depending on the electronic band structure of CdTe NC thin film (with valance band ~5.3 eV and work function of ~5.2 eV), the insertion of an HTL layer with the HOMO level ~5.3 eV can efficiently enhance hole collecting efficiency and form an ohmic contact by band alignment engineering between the back contact electrode and the CdTe NC thin film. In 2015, Yang et al. first developed a commercially available Spiro as the HTL for CdTe NC solar cells [27]. It was found that the surface potential between CdTe and HTL is 80 mV more positive than that of Spiro. A dipole layer can be formed between CdTe and HTL and reinforces the built-in field (*V*_bi_), leading to a higher *V*_oc_ in this case. By optimizing the Spiro HTL, a PCE as high as 8.34% was obtained for CdTe NC solar cells [28]. Later on, a cross-linkable conjugated polymer poly(diphenylsilane–co–4–vinyl–triphenylamine) (Si-TPA) was adopted as HTL for CdTe NC solar cells. A high PCE of 8.34% was obtained due to low contact resistance as a result of good energy level matching [29]. After surface modifications of interface in both cathode and anode, the recombination at interfaces can be greatly decreased, and a PCE of as high as 9.2% was obtained [12]. Although Spiro-OMeTAD is one of the most efficient HTLs for PSCs, it must be chemically doped with Li salt or other composites to obtain an optimal hole collecting efficiency and conductivity. The dopant will badly influence the stability of solar cells since Li+ or other metal ions may diffuse into the active layer as the recombining center [30,31,32]. Searching for a candidate to replace Spiro-OMeTAD is a promising way to solve the problem. Among all the alternatives, PTAA, a conjugated polymer, shows similar HOMO and LUMO level as that of Spiro-OMeTAD. PTAA has been successfully applied in PSC as HTL in recent years [33]. However, there are still no reports on the application of this HTL to CdTe NC solar cells. Herein, we first introduce PTAA as HTL into CdTe NC solar cells and a photovoltaic device with the configuration of ITO/ZnO/CdS(w/o)/CdSe/CdTe/HTL/Au is successfully fabricated. It was found that the PTAA concentration and annealing temperature have a significant effect on the device performance. The optimized device shows an open circuit voltage (*V*_oc_), short circuit current density (*J*_sc_), fill factor (FF) and power conversion efficiency (PCE) of 0. 71 V, 23.11 mA/cm^2^, 50.83% and 6.95%, respectively, which is a significant improvement compared to the control device without it (PCE = 5.3%). Moreover, the NCs solar cells (uncapped) show excellent stability with less than 3% decay during 30 days of measurement.

## 2. Experiment

*Device Fabrication*: CdS NC, CdSe NC, CdTe NC and ZnO precursors were prepared based on previously reported methods [34,35,36,37]. PTAA (CAS: 1333317-99-9) was purchased from Derthon Optoelectronic Materials Science Technology Co., Ltd. (Shenzhen, China) and used as received. In a typical device fabrication process, as shown in Figure 1, ZnO precursor was spin–coated on the ITO substrates at a speed of 3000 rpm for 20 s under ambient conditions. After that, the sample was annealed at 200 °C for 10 min to form a ZnO thin film, and then annealed at 400 °C for 10 min to eliminate the impurities. Subsequently, CdS, CdSe and CdTe NC thin films were deposited through layer-by-layer spin-coating process under ambient conditions. Details can be found in the previous published literature [34,35]. As for the fabrication of PTAA HTL, PTAA was dissolved in chlorobenzene with a concentration of 3, 5 and 7 mg/mL. The PTAA solution was dropped onto CdTe NC film and spun at 2000 rpm for 20 s. Then the sample was annealed at 120 °C for 10 min. An 80 nm thick gold electrode with an active area of 0.16 cm^2^ was deposited via vacuum thermal evaporation through a shadow mask (Appendix A).

*Characterization*: UV-vis absorption characterization was carried out using a Shimadzu 3600 UV-vis-NIR spectrophotometer. The electrochemical cyclic voltammetry was conducted using a CHI660E Electrochemical Workstation. Atomic force microscopy (AFM) imagines were obtained using a NanoScope NS3A system (Veeco, San Jose, CA, USA). X-ray photoelectron spectra (XPS) were measured using Thermo Fisher Scientific’s K-ALPHA+. The SCLC measurement and *J*–*V* characteristics under dark and 100 mW·cm^−2^ (AM 1.5 G) illumination condition were performed using a solar simulator (Oriel model 91192). EQE measurements were recorded utilizing a spectral response/quantum efficiency measurement testing system (QE–R3011, Enlitech, Kaohsiung, Taiwan). The *C*–*V* characteristic of the device was measured by an impedance analyzer (HP-4192A, Hewlett Packard, Palo Alto, CA, USA) under a dark environment at 1000 Hz from −1.0 to 1.0 V.

## 3. Results

Figure 2a shows the absorption spectrum of PTAA, which has a strong single absorption peak at 421 nm, indicating a bandgap of about 2.94 eV. The cyclic voltammetry curve of PTAA in chlorobenzene solution is presented in Figure 2b. Based on $E_{HOMO}=-(E_{ox.,onset}+4.4\;\mathrm{eV}$) and the oxidation potential ($E_{ox.,onset}$) of PTAA (0.8 eV), the highest occupied molecular orbital (HOMO) of −5.20 eV for PTAA was obtained with the corresponding lowest occupied molecular orbital (LUMO) level of −2.26 eV. The HOMO value of PTAA was close to the work function of CdTe NC thin film (~5.17 eV [12]), making PTAA promising for compatible band alignment between CdTe NC film and contact electrode.

A smooth and compact interface could help decrease defects at the semiconductor/metal contacts and ensure efficient carrier extraction. To optimize the interface morphology of the semiconductor/back contact electrode in our NC devices, the PTAA layer with different experimental conditions (using PTAA precursor with different concentrations) was deposited onto ITO/CdTe NC (two layers) and annealed at 110 °C for 10 min under ambient condition. From Appendix A, we could see that the contact angle of PTAA solution on CdTe NC thin film becomes small, indicating a good wetting condition of PTAA solution on CdTe NC thin film. AFM surface morphology of different samples are shown in Figure 3a–d. All samples show a very smooth surface with root-mean-squares (RMS) value below 12 nm (the corresponding RMS values are 6.64 nm, 11.2 nm, 6.71 nm and 5.67 nm, respectively, for 0, 3, 5 and 7 mg/mL PTAA precursors). It is noted that even at high PTAA concentration, the RMS value is low, which implies the high quality PTAA film formed on the CdTe NC surface.

The thickness of HTL is important for optimizing the architecture parameters of solar cells. If the HTL is too thin, it is difficult to form an effective junction when coupling with the light absorption layer. As a result, the HTL is less effective for promoting carrier extraction rate. On the other hand, if the HTL is too thick, the parasitic resistance of the device will increase and thus harm the device performance. We further tested the performance of the device with different thicknesses of PTAA. All the devices with the same architecture of ITO/ZnO/CdSe/CdTe/HTL/Au were prepared at the same conditions except that the HTL thickness used different concentrations of PTAA solution. The structure of the control device A was ITO/CdSe/CdTe/Au. The structure of device B, C and D was ITO/CdSe/CdTe/PTAA/Au where the PTAA concentration was 3, 5 and 7 mg/mL, respectively. The structure of devices E, F and G was ITO/CdSe/CdTe/PTAA (5 mg/mL)/Au where the PTAA annealing temperature was 100, 120 and 140 °C, Device H was ITO/ZnO/CdS/CdSe/CdTe/Au and Device I is ITO/ZnO/CdS/CdSe/CdTe/PTAA (5 mg/mL) with 120 °C annealing/Au, respectively. Figure 4a exhibits the current density–voltage (*J*–*V*) curves of the NC solar cells with or without PTAA HTL, and Figure 4b shows the *J*–*V* curves of devices with different PTAA HTL annealing temperatures. The device performance is summarized in Table 1. The control device without HTL showed a *V*_oc_ of 0.58 V, short circuit current density (*J*_sc_) of 14.61 mA·cm^−2^ and a fill factor (FF) of 0.42, delivering a low PCE of 3.58%. In contrast, with the PTAA concentration increase from 3 to 7 mg mL, device performances were effectively enhanced. At the PTAA concentration of 5 mg/mL, we obtained the champion device which showed a *J*_sc_ of 21.06 mA·cm^−2^, a high *V*_oc_ of 0.68 V and a FF of 0.37, resulting in a high PCE of 5.41%. Obviously, the *J*_sc_ and *V*_oc_ of devices with PTAA HTL are simultaneously improved. The PCE of a device with PTAA is over 50% higher than the control device. From Table 1, one can see that the improvement of efficiency in NC solar cells with PTAA HTL is mainly attributed to the increase in *V*_oc_ (from 0.58 to ~0.68 V) and *J*_sc_ (from 14.61 to 21.06 mA/cm^2^). As a result of the high-quality diode, the photo-induced electron-hole pair could be effectively separated by the built-in electric field (*V*_bi_). Combining the data shown in Figure 4a and Table 1, it can be seen that the optimized PTAA solution concentration is located at 3~5 mg/mL. Lower concentration of PTAA may result in inadequate coverage of the CdTe thin film surface, and higher concentration of PTAA may lead to large parasitic resistance which increases the *R*_s_ of the device.

In the previous reports, it has been found that the HTL annealing temperature also has a great effect on the device performance. In this part, all the devices have the same architecture of ITO/ZnO/CdSe/CdTe/HTL/Au with the same thickness of HTL while the annealing temperature is different (from 25 °C to 140 °C). The *J*–*V* curves of devices with different annealing temperatures under AM 1.5 G illumination condition are presented in Figure 4b, while the device parameters are listed in Table 1. It is shown that the PCE of devices increases with the increase of annealing temperature from 25 °C to 120 °C and starts to drop when the annealing temperature exceeds 120 °C. The champion device is obtained when annealing at 120 °C. The optimized device shows a *J*_sc_ of 22.33 mA/cm^2^, a *V*_oc_ of 0.69 V, an FF of 37.17% and a high PCE of 5.70%. The optimal PCE is 30% higher than the control device. For devices annealed at low temperature, a serious interface recombination is expected since the HTL does not fully passivate the CdTe NC thin film surface, and the HTL has abundant intrinsic defects. On the contrary, when the HTL annealing temperature is too high, the oxidation of PTAA may take place, leading to a larger *R*_s_ and lower PCE. It is noted that the FF of devices with HTL is somewhat lower than that of a device without HTL. We speculate that although the introduction of PTAA can reduce the interface carriers recombination, the low hole concentration of PTAA leads to large contact resistance, which decreases the FF of the device. The FF of the device can be further improved by doping the PTAA with inorganic salt as similar results have been obtained in the case of PSCs with Spiro as HTL.

In order to further optimize the electron/hole transfer balance and improve the NC device performance, we fabricated bilayer CdS electron transport layer–based (ETL) devices with the configuration of ITO/ZnO/CdS/CdSe/CdTe/HTL(w/o)/Au, which consists of one layer of CdS NC, two layers of CdSe NC and five layers of CdTe NC. Figure 5a shows the *J*–*V* curves of NC solar cells under AM 1.5 G, with the corresponding EQE spectrum and *J*–*V* under dark presented in Figure 5b,c. The control device without HTL shows a *V*_oc_ of 0.62 V, *J*_sc_ of 15.41 mA cm^−2^ and FF of 0.56, delivering a PCE of 5.34%. In contrast, the device with PTAA shows a *V*_oc_ of 0.69 V, *J*_sc_ of 21.47 mA cm^−2^ and FF of 0.48, delivering a much higher PCE of 6.95%. The PCE is increased by 35% when HTL is introduced. From the EQE spectra (Figure 5b), one can gain more insights into how the *J*_sc_ varies as a function of the wavelength of incident light. The device with HTL annealing at optimal temperature of 120 °C exhibits a higher EQE value almost in the whole wavelength, suggesting the improved carrier collecting efficiency due to the reduced interface defects between the CdTe NC layer and the Au electrode. When the EQE curves are integrated with the AM1.5G spectrum, current densities of 15.14 mA/cm^2^ and 22.32 mA/cm^2^ are obtained from measured EQE curves, respectively, which is consistent with our *J*–*V* measurement (Figure 5a). The dark *J*–*V* characteristics of NC solar cells are shown in Figure 5c. At reverse bias, the leakage current is decreased almost one-fold compared to the device without HTL, indicating the importance of CdS ETL and PTAA HTL for further optimizing device performance.

To understand the origin of the improved *J*_sc_ for devices with HTL, the hole-only devices with ITO/CdTe (200 nm)/PTAA (20 nm, w/o)/Au configuration are fabricated. The hole mobility is measured by space charge limited current (SCLC). The hole mobility of NC films are calculated based on the Mott–Gurney law [38]:$$J=\frac{9}{8}\varepsilon_{0}\varepsilon_{r}\mu_{p}\frac{{\left(V-V_{\mathrm{bi}}-V_{s}\right)}^{2}}{L^{3}}$$
where *L* is the thickness of the NC/PTAA layer, $\varepsilon_{0}$ is the permittivity of free space, $\varepsilon_{r}$ is the relative dielectric constant of CdTe (9.8), μ_p_ is the hole mobility, *V* is the applied voltage, and *V*_s_ is the voltage drop due to contact resistance, while *V*_bi_ is the built-in voltage. From Figure 6a,b, the mobility for a device with and without PTAA are 7.76 × 10^−4^ and 3.07 × 10^−4^ cm^2^/Vs, respectively. The μ_p_ is increased by a factor of 2–3 with the presence of HTL. The higher mobility for the device with PTAA implies that hole carrier transport is improved, and thus a high *J*_sc_ is expected for NC devices with optimal HTL thickness and annealing temperature.

To explain the functional mechanism of interaction between PTAA and the CdTe NC layer, ITO/CdTe NCs (2 layers)/PTAA (2 nm, annealing at 120 °C) samples are fabricated and investigated by X-ray photoelectron spectroscopy (XPS) spectra. Figure 7a shows the XPS narrow scan of Cd 3d, Te 3d and N 1s for thin films with and without PTAA HTL. The binding energies of the Cd 3d, Te 3d and N 1s are summarized in Table 2. Appendix A shows the full scan spectra of the thin film. The presence of an N 1s signal is attributed to the elements of PTAA [39]. Compared to the control sample without PTAA HTL (The peak location are 405.72 eV and 412.60 eV, respectively), one can see that the binding energies of Cd 3d_5/2_ and Cd 3d_3/2_ are located at 405.13 eV and 412.01 eV, respectively, for the sample with PTAA HTL. The binding energies of the Cd 3d core level with PTAA coverage show a 0.59 eV higher than the control sample. We speculated that the existence of Cd/N bonding should attract electron from Cd to N, and a dipole layer can be formed between the CdTe NC layer and HTL. Therefore, the binding energies of the Cd 3d is increased. Similar results were reported in the previous work with Spiro as HTL [40,41]. As the direction of the dipole moment is aligned with the native *V*_bi_, the new *V*_bi_ in solar cells will be reinforced, which facilitates hole transport/separation on the surface of CdTe NC and contributes to a higher *V*_oc_ (Figure 5a). To confirm the improved built-in electric field of CdTe NC solar cells with PTAA HTL, capacitance–voltage analysis is carried out with the increase of bias voltage at a constant frequency of 1000 Hz. According to the Mott–Schottky equation [42]:$$C^{-2}=\frac{2}{A^{2}q\varepsilon_{0}\varepsilon N_{A}}\left(V_{\mathrm{bi}}-V\right)\;$$
where *A* is the device area, *ε* is the relative dielectric constant (10.6), *ε*_0_ is the permittivity of vacuum, *V*_bi_ the built-in potential, and *N*_A_ is the net acceptor concentration. The *C*^−^^2^-*V* curves are presented in Figure 7b, and the *V*_bi_ can be extracted at a forward bias from the slope (the extrapolated intersection with the *x* axis). The *V*_bi_ of the NC device with PTAA is 0.69 V, while this value is only 0.62 V for the control devices, which agrees well with their *V*_oc_ values from the *J*–*V* curves. Based on the experiments results, the band alignment and interaction between CdTe NC and PTAA are shown in Figure 7c. We speculate that when PTAA is deposited on the ITO/ZnO/CdSe/CdTe substrate, a dipole layer is formed between the CdTe NC layer and PTAA (by forming a Cd-N bond). A similar result has been found in the case of CdTe NC solar cells with Spiro as HTL. As the direction of the dipole field (from CdTe to HTL) is the same as the built-in potential (from ZnO to CdTe), the overall built–in field will be reinforced, leading to high *V*_oc_. Similar to the effects of P-TPA HTL [12], the PTAA should passivate the interface of CdTe NC layer and reduce electron/hole recombination at the CdTe/Au interface, resulting in better device performance.

## 4. Conclusions

In conclusion, we have demonstrated a commercial polymer PTAA as a good alternative of HTLs in solution–processed CdTe NC solar cells. By optimizing the HTL thickness and annealing temperature, the open circuit voltage, fill factor and short circuit current of NC solar cells are effectively promoted, resulting in a high PCE of 6.95%, which is increased by 30% when comparing to a control device without PTAA. The improvement in binding energy of Cd 3d and *V*_bi_ suggests that a dipole layer is formed between CdTe and PTAA, which increases carrier collection efficiency and reduces interface recombination. This work presents a new way for further improving the performance of a solution-processed NC device by incorporating a suitable conjugated polymer as the HTL.

## Figures and Tables

**Figure 1 nanomaterials-12-03067-f001:**
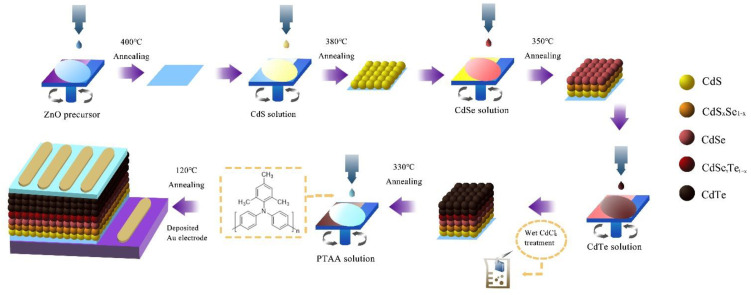
A schematic of NC solar cell fabrication process.

**Figure 2 nanomaterials-12-03067-f002:**
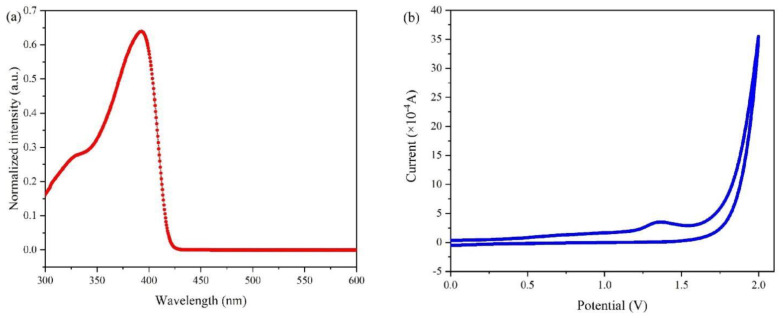
(**a**) UV−vis absorption spectrum of PTAA in chlorobenzene solution; and (**b**) cyclic voltammetry curve of PTAA in chlorobenzene solution.

**Figure 3 nanomaterials-12-03067-f003:**
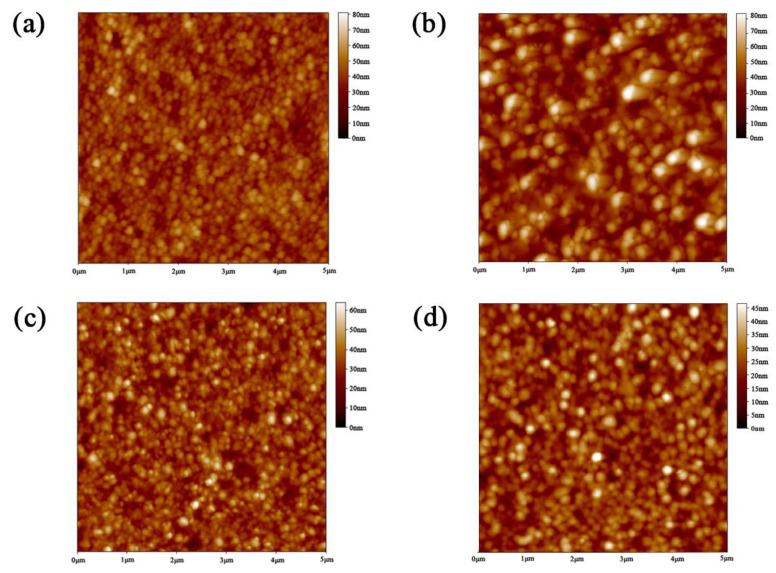
Atomic force microscopy (AFM) of PTAA on ITO/ZnO/CdS/CdSe/CdTe with different concentrations: (**a**) without PTAA; (**b**) 3 mg mL^−1^ PTAA; (**c**) 5 mg mL^−1^ PTAA; (**d**) 7 mg mL^−1^ PTAA.

**Figure 4 nanomaterials-12-03067-f004:**
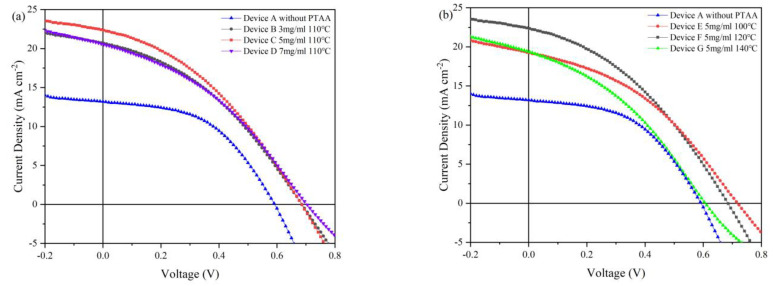
*J*–*V* characteristics of ITO/ZnO/CdSe/CdTe/PTAA/Au devices with (**a**) different PTAA concentrations and (**b**) annealing temperatures.

**Figure 5 nanomaterials-12-03067-f005:**
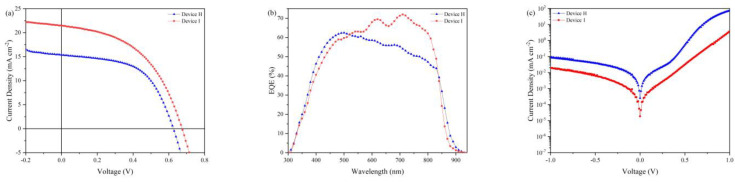
(**a**) *J*–*V* curves of ITO/ZnO/CdSe/CdTe/PTAA(w/o)/Au. (**b**) Corresponding EQE spectra. (**c**) Dark *J*–*V* characteristic curves.

**Figure 6 nanomaterials-12-03067-f006:**
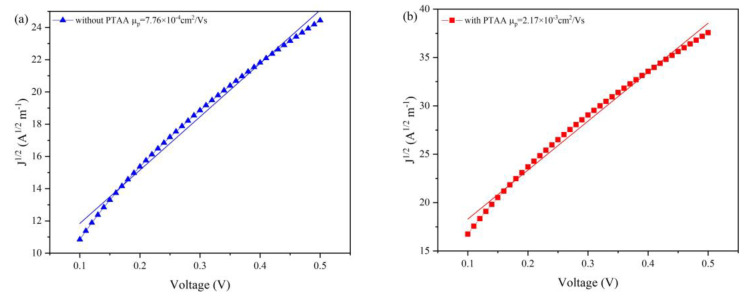
SCLC measurements of holy−only devices: (**a**) without; and (**b**) with PTAA (device structure: ITO/CdTe (200 nm)/PTAA/Au.

**Figure 7 nanomaterials-12-03067-f007:**
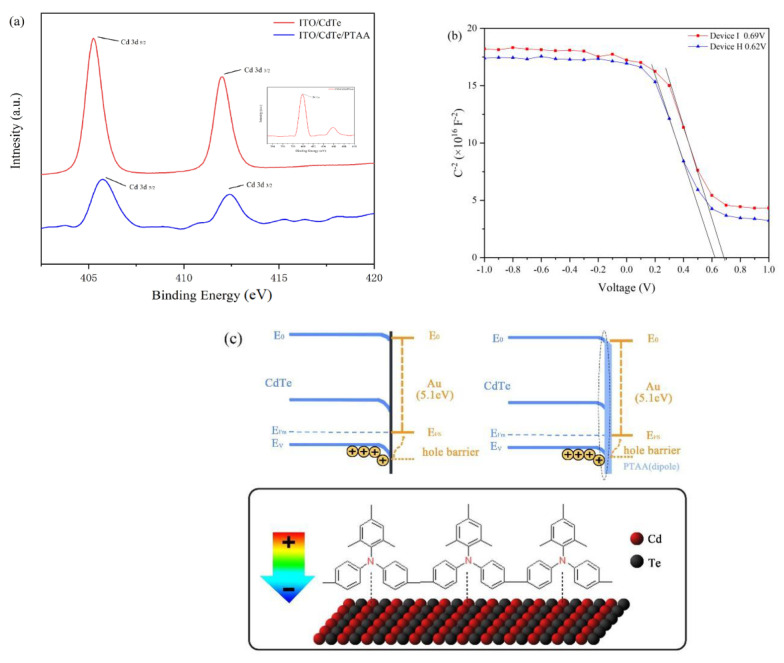
(**a**) The XPS spectra of Cd 3d peaks; (**b**) Mott–Schottky C−V curve of devices; (**c**) schematic of energy levels influenced by a dipole layer (PTAA) at the CdTe/Au interface.

**Table 1 nanomaterials-12-03067-t001:** Summarized photovoltaic parameters from *J*–*V* curves of CdTe NC solar cells prepared under different conditions.

Device	V_OC_ (V)	J_SC_ (mA cm^−2^)	FF (%)	PCE (%)
A	0.58	14.61	42.28	3.58
B	0.67	20.32	36.87	5.00
C	0.68	21.06	37.40	5.41
D	0.70	21.54	35.38	5.29
E	0.71	19.26	39.26	5.40
F	0.69	22.33	37.17	5.70
G	0.60	19.35	36.44	4.25
H	0.62	15.41	55.96	5.34
I	0.68	21.47	47.81	6.95

**Table 2 nanomaterials-12-03067-t002:** Binding energy of N, P and Cd elements in CdTe and CdTe/PTAA films.

Thin Film	N 1s	P 2p	Cd 3d_3/2_	Cd 3d_5/2_
CdTe	-	133.12	412.01	405.13
CdTe/PTAA	399.82	133.71	412.60	405.72

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
