# Peer review of "Efficient Nanocrystal Photovoltaics with PTAA as Hole Transport Layer"

_nanomaterials, 2022, doi:10.3390/nano12173067_

Round 1
Reviewer 1 Report
I congratulate the authors, for my part, the article is completely acceptable in its present form. I have only a few formal suggestions that do not require scientific correction, so I will refrain from listing them.
Author Response
Reviewer #1:
I congratulate the authors, for my part, the article is completely acceptable in its present form. I have only a few formal suggestions that do not require scientific correction, so I will refrain from listing them.
Thanks for the comments. We further improve the quality of this manuscript in the revised manuscript.
Reviewer 2 Report
In this work, the authors demonstrate PTAA as an alternative HTL for CdSe NC solar cells. Simultaneous improvements in Jsc, Voc and FF were obtained upon modifying the hole selective electrode with the incorporation of an optimized thickness PTAA layer. Various experiments were contacted to uncover the origins of the improved device performance and reasonable explanations are provided to this regard. Apart a significant improvement of the English language and grammar used throughout the manuscript upon editing by someone with a very good knowledge of written English, some revisions should be undertaken to clarify specific issues arising and improve the manuscript prior to reconsideration for publication.
Below, a list of my comments to the authors is provided.
1. In the introduction, the authors refer to the term PSCs without giving its full name before (I presume they refer to perovskite solar cells). Also, while they first discuss CdSe nanocrystal solar cells, they then discuss the role of various HTLs in perovskite solar cells and then they start to discuss again various HTLs employed ih nanocrystal solar cells and their limitations as well as their roles in enhnacing solar cell performance. This is somewhat confusing to the reader and that part of the introduction should be rewritten in a more cohesive manner that will guide the reader in what has been done and what is proposed by the authors in their work.
2. Some figures such as Figs. 4 and 5 are of low quality and should be definitely improved.
3. The authors note in the introduction and the main part of the manuscript that a significant impovement of the FF is obtained upon incorporating PTAA as a HTL. Hovewer, this is not supported by data presented in Fig. 4 and Table 1 where the device FF decreases upon incorporation of the PTAA HTL as well as upon its annealing (devices B to G have lower FF than the reference device A). The authors should discuss this finding in more detail.
4. In page 7, it is mentioned that the hole mobility increases one order of magnitude with the presence of the PTAA HTL. However, the given values are 7.76×10-4 and 3.07×10-4 cm2/Vs which is an increase of a factor of 2-3 (not an order of magnitude as claimed).
5. Concerning the XPS spectra, the shift of the Cd 3d peaks to lower binding energies upon incorporating the PTAA HTL should be more clearly explained. The authors speculate on the existence of Cd/N bonding that will attract electron from Cd to N. Is the shift to lower binding energies thus attributed to electron transfer ? Moreover, the formation of the interface dipole layer between CdTe and PTAA is attributed to Cd-N bonding by the authors. Could this be due to spontaneous electron transfer upon contacting PTAA with CdTe, due to their different Fermi levels rather than chemical bonding ? This dipole could be probed by UPS spectroscopy and verified if its value agrees with the difference in the Vbi as measured by C-V.
6. In Page 8, the authors refer to PTAA as passivator of the interface of CdTe NC layer which reduces electron/hole recombination at the CdTe/Au interface noting Figure7b. However, no specific evidence is provided for the possible role of PTAA as a passivation layer. Figure 7b, demonstrating an enhanced Vbi, can not be used to verify that claim. PL and TRPL spectroscopy could be used to possibly support the role of PTAA as a passivation layer at the CdSe/metal interface.
Author Response
Response to the referee’s report
nanomaterials- 1870861
(Italic character- Referee’s comment; red character-author’s response to the referee)
Reviewer #2:
In this work, the authors demonstrate PTAA as an alternative HTL for CdSe NC solar cells. Simultaneous improvements in Jsc, Voc and FF were obtained upon modifying the hole selective electrode with the incorporation of an optimized thickness PTAA layer. Various experiments were contacted to uncover the origins of the improved device performance and reasonable explanations are provided to this regard. Apart a significant improvement of the English language and grammar used throughout the manuscript upon editing by someone with a very good knowledge of written English, some revisions should be undertaken to clarify specific issues arising and improve the manuscript prior to reconsideration for publication.
Below, a list of my comments to the authors is provided.
- In the introduction, the authors refer to the term PSCs without giving its full name before (I presume they refer to perovskite solar cells). Also, while they first discuss CdSe nanocrystal solar cells, they then discuss the role of various HTLs in perovskite solar cells and then they start to discuss again various HTLs employed ih nanocrystal solar cells and their limitations as well as their roles in enhnacing solar cell performance. This is somewhat confusing to the reader and that part of the introduction should be rewritten in a more cohesive manner that will guide the reader in what has been done and what is proposed by the authors in their work.
Thanks for the comments. The term PSCs is referred to perovskite solar cells. Both the CdTe NC and perovskite are inorganic solar cells and they have similar properties such as the work function of active layer. Therefore the HTLs used in PSCs can also be applied in CdTe NCs solar cells, which have been confirmed in our previous works. We have rewritten the introduction part in the revised manuscript.
- Some figures such as Figs. 4 and 5 are of low quality and should be definitely improved.
Thanks for the good suggestions. We have improved the quality of Fig. 4 and 5 in the revised manuscript.
- The authors note in the introduction and the main part of the manuscript that a significant impovement of the FF is obtained upon incorporating PTAA as a HTL. Hovewer, this is not supported by data presented in Fig. 4 and Table 1 where the device FF decreases upon incorporation of the PTAA HTL as well as upon its annealing (devices B to G have lower FF than the reference device A). The authors should discuss this finding in more detail.
Thanks for the comments. We have discussed this finding in more detail in the revised manuscript.
- In page 7, it is mentioned that the hole mobility increases one order of magnitude with the presence of the PTAA HTL. However, the given values are 7.76×10-4 and 3.07×10-4 cm2/Vs which is an increase of a factor of 2-3 (not an order of magnitude as claimed).
Thanks for the comments. We have revised this expression in the revised manuscript.
- Concerning the XPS spectra, the shift of the Cd 3d peaks to lower binding energies upon incorporating the PTAA HTL should be more clearly explained. The authors speculate on the existence of Cd/N bonding that will attract electron from Cd to N. Is the shift to lower binding energies thus attributed to electron transfer ? Moreover, the formation of the interface dipole layer between CdTe and PTAA is attributed to Cd-N bonding by the authors. Could this be due to spontaneous electron transfer upon contacting PTAA with CdTe, due to their different Fermi levels rather than chemical bonding ? This dipole could be probed by UPS spectroscopy and verified if its value agrees with the difference in the Vbi as measured by C-V.
Thanks for the good suggestion. If the presence of HTL makes Cd shift to lower binding energies, which will results in the reverse direction of dipole field (when comparing to the direction of build in field) and lead to low Voc. In the previous work (Yang et al, ACS Applied Materials & Interfaces 2016, 8, 900-907, doi:10.1021/acsami.5b10374.), the surface potential images of CdTe NC and Spiro were measured by SKPM(Kelvin probe microscopy). They found that the potential of CdTe NC film showed 80 mV more positive than that of the spiro-OMeTAD, which implied that when spiroOMeTAD serves as HTL, the direction of the dipole moment is aligned with the native Vbi , which will reinforce the ultimate Vbi and enlarge the Voc of the solar cells. As the TPAA has similar HOMO and LUMO value as that of spiro-OMeTAD, it is expected that similar dipole layer is formed between CdTe and TPAA. We have revised the manuscript according to the referee’s suggestion.
- In Page 8, the authors refer to PTAA as passivator of the interface of CdTe NC layer which reduces electron/hole recombination at the CdTe/Au interface noting Figure7b. However, no specific evidence is provided for the possible role of PTAA as a passivation layer. Figure 7b, demonstrating an enhanced Vbi, can not be used to verify that claim. PL and TRPL spectroscopy could be used to possibly support the role of PTAA as a passivation layer at the CdSe/metal interface.
Thanks for the good suggestion. In our previous work (Interface Engineering for Both Cathode and Anode Enables Low-Cost Highly Efficient Solution-Processed CdTe Nanocrystal Solar Cells. Advanced Functional Materials 2019, 29, doi:10.1002/adfm.201904018.), the charge recombination lifetime for device w/o HTL is investigated by transient photovoltage (TPV). We found that the charge recombination recombination lifetime for device with HTL is 1.85µs, which is significantly higher than device without HTL(1.29 µs). As the HOMO value of TPAA is close to that of P-TPA, we speculate that the charge recombination is low for the PTAA processed device. We have revised this expression according to the referee’s suggestion.
Reviewer 3 Report
This manuscript proposes efficient nanocrystal photovoltaics with PTAA as hole transport layer. The topic is interesting, and certainly consistent with the contents to be proposed to the readers of “Nanomaterials”. Moreover, the manuscript is well written and can be read with pleasure: this represents an important aspect in the current scenario of publications in international journals. Overall, I think that this manuscript has to be accepted, but the Authors should take into account the following minor revisions (in terms of bibliographic updates, grammar corrections and content deepening):
- Detailed revisions: I spent several hours reading this manuscript, and Authors are asked to follow carefully the attached PDF file where I highlighted some points to be addressed. The attached file also contains language mistakes and typos; some questions related to manuscript contents could also be present and Authors must consider them properly before submitting the revised manuscript. A point-by-point reply is required when the revised files are submitted.
- The Introduction should give a wider overview on the present scenario related to recent trends in photovoltaics, both in terms of recently published reviews and research articles. In particular, integrated devices and sustainable solutions are missing and a paragraph on this topic is highly suggested to be added in the Introduction. Authors are invited to go through the literature published in the last six months on these issues, and also on concepts developed some years ago in this field. Some of them are also mentioned in the above mentioned PDF file.
- Authors should provide a clear explanation on the experimental error of the proposed research work. In particular, reproducibility of the phenomena described in the manuscript should be clearly stated in the “Results and Discussion” section; besides, some notes in the “Materials and Methods” section should be added highlighting which kind of experimental approach has been followed to check the reproducibility of the proposed system, the latter being of noteworthy importance in the present research field.

Author Response
Reviewer #3:
a good piece of work is presented in this study. HTL was borrowed by the author to improve the performance of CdTe NC solar cells. the work can be published after minor revision
Thanks for the good suggestions. We have revised the manuscript according to the referee’s suggestion.
Reviewer 4 Report
a good piece of work is presented in this study. HTL was borrowed by the author to improve the performance of CdTe NC solar cells. the work can be published after minor revision
Author Response
Reviewer #4:
a good piece of work is presented in this study. HTL was borrowed by the author to improve the performance of CdTe NC solar cells. the work can be published after minor revision.
Thanks for the comments. We have revised the manuscript according to the referee’s suggestion.
Round 2
Reviewer 2 Report
The revisions made by the authors are sufficient and satisfactory. Thus, the manuscript can be accepted in its revised form.